# Influence of age on gadoxetic acid disodium-induced transient respiratory motion artifacts in pediatric liver MRI

**Azadeh Hojreh**[1], **Ahmed Ba-Ssalamah**[1], **Christian Lang**[1,2], **Sarah Poetter-Lang**[1], **Wolf-Dietrich Huber**[3], **Dietmar Tamandl**[1]*

**1** Department of Biomedical Imaging and Image-Guided Therapy, Medical University of Vienna, Vienna, Austria, **2** Department of Anaesthesia, Emergency Medicine and Intensive Care, General Hospital Wiener Neustadt, Wiener Neustadt, Austria, **3** Department of Pediatrics and Adolescent Medicine, Medical University of Vienna, Vienna, Austria

* dietmar.tamandl@meduniwien.ac.at

## Abstract

**Data Availability Statement:** All relevant data are within the paper and its Supporting Information files.

**Funding:** The authors received no specific funding for this work.

### Purpose

Gd-EOB-DTPA-enhanced liver MRI is frequently compromised by transient severe motion artifacts (TSM) in the arterial phase, which limits image interpretation for the detection and differentiation of focal liver lesions and for the recognition of the arterial vasculature before and after liver transplantation. The purpose of this study was to investigate which patient factors affect TSM in children who undergo Gd-EOB-DTPA-enhanced liver MRI and whether younger children are affected as much as adolescents.

### Methods

One hundred and forty-eight patients (65 female, 83 male, 0.1–18.9 years old), who underwent 226 Gd-EOB-DTPA-enhanced MRIs were included retrospectively in this single-center study. The occurrence of TSM was assessed by three readers using a four-point Likert scale. The relation to age, gender, body mass index, indication for MRI, requirement for sedation, and MR repetition was investigated using uni- and multivariate logistic regression analysis.

### Results

In Gd-EOB-DTPA-enhanced MRIs, TSM occurred in 24 examinations (10.6%). Patients with TSM were significantly older than patients without TSM (median 14.3 years; range 10.1–18.1 vs. 12.4 years; range 0.1–18.9, p<0.001). TSM never appeared under sedation. Thirty of 50 scans in patients younger than 10 years were without sedation. TSM were not observed in non-sedated patients younger than 10 years of age (p = 0.028). In a logistic regression analysis, age remained the only cofactor independently associated with the occurrence of TSM (hazard ratio 9.152, p = 0.049).

**Competing interests:** The authors have declared that no competing interests exist.

**Abbreviations:** AIH, Autoimmune hepatitis; B, Unstandardized beta; CI, Confidence interval; FA, Flip angle; FOV, Field of view; HR, Hazard ratios; IBD, Inflammatory bowel disease; IRB, Institutional Review Board; GBCA, Gadolinium-based contrast agent; Gd-EOB-DTPA, Gadoxetate disodium; GRE, Gradient echo; GvHD, Graft versus host disease; IQ, Image quality; PBC, Primary biliary cholangitis; PT, Patients; PSC, Primary sclerosing cholangitis; SL, Slice thickness; SpO2, Peripheral capillary oxygen saturation; TE, Echo time; TR, Pulse repetition time; TSM, Transient severe motion; VIBE, Volumetric interpolated breath-hold examination.

## Conclusion

TSM in Gd-EOB-DTPA-enhanced liver MRI do not appear in children under the age of 10 years.

## Introduction

MRI is the preferred imaging modality for the assessment of liver pathologies in daily clinical practice, both for adults, and also, increasingly, for pediatric patients [1, 2]. Gadoxetate disodium (Gd-EOB-DTPA, Primovist® or Eovist®) has improved liver magnetic resonance imaging due to its superior performance in diffuse and focal liver diseases [3–8], with the benefit of not only superior lesion detection and characterization, but also the possibility to perform an assessment of liver function [3, 9], which is also applicable in children [7, 10, 11]. For the detection and differentiation of focal liver lesions, the image quality of the arterial phase in Gd-EOB-DTPA-enhanced liver MRI is essential [12–14], and proper assessment of the arterial vasculature is also required for patients before and after liver transplantation [15–17]. However, the image quality of Gd-EOB-DTPA-enhanced liver MRI is sometimes limited by transient severe motion (TSM) artifacts in the arterial phase, which are the result of acute transient dyspnea and reduced breath-holding capacity [18–20]. Probable causes and suggested solutions have been discussed controversially over the last several years in adult patients, but have been less thoroughly studied in children [3, 20–26]. Thus far, it is known from pediatric patients that sedation appears to have a protective effect against TSM, yet other confounders, such as age, have not been studied in detail as yet [27, 28].

The purpose of this study was to investigate which patient factors affect TSM after the administration of Gd-EOB-DPTA in pediatric patients, and whether younger children are affected as much as adolescents.

## Materials and methods

### Patients and clinical data

The Institutional Review Board of Medical University of Vienna (IRB No. 1296/2017) approved this retrospective, single-center study, and written, informed consent was waived for the data analysis. All procedures performed in the study that involved human participants were in accordance with the ethical standards of the institutional review board and with the 1964 Helsinki declaration and its later amendments.

All pediatric Gd-EOB-DTPA-enhanced liver MRI scans performed at our tertiary referral center between July 2012 and March 2017 were included in this study.

Inclusion criteria were patient age younger than 19 years and examination performed on a single MRI scanner (1.5 Tesla Magnetom AERA®, Siemens Healthineers, Siemens Healthcare GmbH), with adherence to the standard liver MRI-examination protocol using Gd-EOB-DTPA (Primovist®, Eovist®, Bayer Vital GmbH) as a contrast agent.

Exclusion criteria were a protocol deviation from the standard liver MRI protocol, the application of a different GBCA other than Gd-EOB-DTPA, and overall non-diagnostic image quality. The selection process is illustrated in Fig 1.

The following clinical data were retrieved from the hospital information system and/or scan documentation: patient age; gender; indication for liver MRI; height; weight; body mass index; and requirement for sedation.

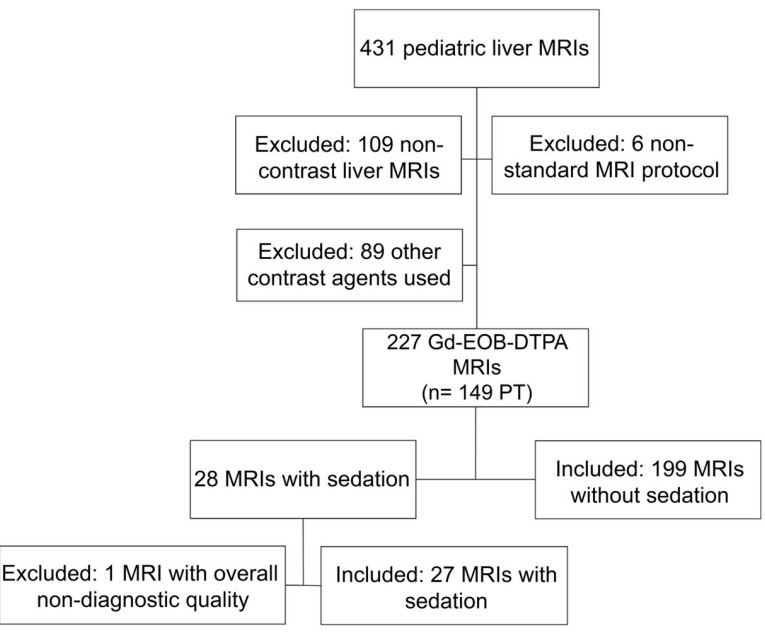

**Fig 1. Flowchart of the included MRIs.**

## MRI examination protocols

The liver MRI examination protocol is described in S1 Table. Contrast administration was performed based on the clinical indication. For the administration of contrast agent, dedicated, informed consent was obtained from the referring physician and the patients' legal guardian.

Need for sedation was decided by the treating radiologist together with a pediatric anesthesiologist. Children were not intubated, but breathed freely with an oxygen mask. All non-sedated children performed breath-hold-commands exercises with our radiology technologists before they went into the MR-scanner.

Gd-EOB-DTPA was applied at a standard dose of 0.025 mmol/kg body weight and was slowly administered manually as an intravenous bolus injection.

The dynamic images were obtained with the same parameters used for the unenhanced sequence with a sequential k-space ordering. The acquisition times were 3 times à 16 seconds, beginning at the time to aortic peak (arterial) and 48 seconds (portal venous), and 300 seconds (equilibrium) (S1 Table).

## Image assessment and interpretation

Three radiologists, including a pediatric radiologist and two abdominal radiologists, with 15, 10, and six years of experience, respectively, in the interpretation of abdominal MRI, read the studies, which were viewed on a PACS (Picture Archiving and Communication System, Agfa) workstation. The readers were blinded to the clinical course of the patient.

Image quality was determined using a four-point Likert scale, modified according to Polanec et al. [3]:

"0" for severe motion artifacts, non-diagnostic image quality;

"1" for marked motion artifacts, reduced diagnostic image quality, but acceptable;

"2" for minimal motion artifacts, good diagnostic image quality; and

"3" for no motion artifacts, excellent diagnostic image quality.

For the arterial phase, the decision criterion was the visibility of the common hepatic artery and the proper hepatic artery, and, for the portal-venous phase and the transitional phases, the decision criterion was the visibility of the intra- and extrahepatic parts of the hepatic portal vein, as well as the liver parenchyma. The hepatobiliary phase was not assessed for the purpose of this study. The mean value of image quality for all three readers was used for further calculations after inter-reader variability was confirmed to be comparable to other studies in this field [28].

The presence of transient severe motion artifacts (TSM) was defined as a decrease of image quality by at least one point in the arterial phase compared to the unenhanced scan, which returned to baseline values until the transitional phase. Patients with non-diagnostic images in all exam phases and/or missing variables (n = 1) were not included in the TSM analysis.

## Statistical analysis

The statistical analysis was performed using the software package SPSS, Version 25.0 (IBM, Armonk, NY). Demographic data were analyzed as absolute numbers, and distributions were presented as median and range, or mean +/- standard deviation, where applicable. Variables were compared using a parametric (Student´s t-test) or non-parametric test (Mann-Whitney U), or the Chi-square test, where applicable.

Inter-reader agreement among the three readers was assessed using κ coefficients.

Logistic regression analysis by age, gender, weight, body mass index (BMI), and sedation, with regard to the occurrence of TSM, was performed. Results are presented as hazard ratios (HR) and 95% confidence intervals (CI). All variables used in the univariate analysis (except BMI due to the collinearity with weight) were included into a multivariate regression model with stepwise backward elimination. Results with a p value <0.05 were considered significant.

## Results

### Patients

A total of 148 patients with 226 Gd-EOB-DTPA-enhanced liver MRI examinations (65 female, 83 male, 0.1–18.9 years old) met the inclusion criteria. The indications for performing a liver MRI were quite variable; however, four general indication groups were identified (transplantation, liver mass, pancreaticobiliary disorders, and hepatobiliary functional liver diseases, Table 1, and S2 Table).

**Assessment of image artifacts—inter-reader agreement.**   The overall inter-reader agreement was substantial (intraclass correlation coefficient: 0.608–0.699); therefore, a mean score for image quality was used, as described above.

**Image quality in the dynamic contrast phases.**   The mean image quality in the unenhanced, arterial, portal venous, and transitional phases was 2.05±0.70, 1.98±0.65, 2.12±0.66, and 2.18±0.65, respectively, with the quality of the arterial phase significantly lower than that of the portal and transitional phases, if TSM were present (p<0.001). This is illustrated in Fig 2A and 2B.

**Assessment of TSM in relation to age and sedation.**   TSM in the arterial phase occurred in 24 examinations (10.6%) in 23 patients after Gd-EOB-DTPA. Patients with TSM were significantly older compared to patients without TSM (median 14.3 years [10.1–18.1] vs. 12.4 years [0.1–18.9], p<0.001). TSM were not observed in patients below the age of 10 years, Fig 3. Weight and BMI was not different between patients with or without TSM (p = 0.062 and 0.393).

Of 226 liver MRIs performed with Gd-EOB-DTPA, 27 (12%) examinations required sedation. No TSM were noted in patients who had MRI under sedation (p = 0.039 vs. exams

**Table 1. Demographics.**

| Characteristics | No. of Gd-EOB-DTPA MRIs |
|---|---|
| No. Patients (%) | 148 |
| • Females (%) | 65 (43.9) |
| Examinations | 226 |
| Patients with a single MRI | 111 |
| Patients with repeated MRIs (%) | 37 (25) |
| • Repeated MRIs per patients | 2 (2–12) |
| Sedation (%) | 27 (11.9) |
| Median age (all patients, y) | 12.6 (0.1–18.9) |
| • With sedation | 4.3 (0.1–16.8) |
| • Without sedation | 13.4 (0.6–18.9) |
| Median weight (all patients, kg) | 45 (2.5–147) |
| BMI (all patients, kg/m$^2$) | 19.2 (8.2–44.1) |
| Indication for liver MRI | |
| • Transplantation | 23 (10.2) |
| • Masses | 71 (31.4) |
| • Pancreaticobiliary disorders | 49 (21.7) |
| • Hepatobiliary functional disorders | 83 (36.7) |

Data presented are absolute numbers or median and range in parentheses, when applicable. Percentages apply to the respective groups, or to the entire cohort

without sedation). Patients who required sedation were younger (median age 4.3 years [0.1–16.8] vs. 13.4 years [0.6–18.9], p<0.001) and of lower median weight and BMI compared to those without sedation (weight 17.0 kg [4.0–75.0] vs. 47.0 kg [2.5–147.0], p<0.001; BMI 16.1 [8.1–24.0] vs. 19.5 [11.3–44.1] kg/m$^2$, p<0.001), Table 1.

Although patients who required sedation were generally younger (Table 1), there was also a considerable number of patients younger than 10 years who did not require sedation. Of the 50 studies in patients younger than 10 years, 30 (60.0%) did not require sedation, while 169 of 176 (96.0%) studies in patients older than 10 years were performed without sedation. The rate of TSM events in the older, non-sedated group was 14.2% (24 of 169 studies), while it was 0.0% (0 of 30 studies) in the younger, non-sedated group (p = 0.028). This is also illustrated in Fig 3, with age as a continuous variable. Fig 4 presents an example of TSM.

**Assessment of TSM in patients undergoing repeated examinations.** Of 37 patients who had repeated Gd-EOB-DTPA-enhanced MRIs, 28 patients did not develop a TSM, and nine patients had at least one TSM event (S3 Table). Five of nine patients with repeated Gd-EOB-DTPA-enhanced MRI had at least one uneventful subsequent Gd-EOB-DTPA-enhanced MRI, after a TSM event. Fourteen patients of 23 patients with TSM events had only a single Gd-EOB-DTPA-enhanced MRI. Only one patient with six Gd-EOB-DTPA-enhanced MRIs had two events, at the first and at the third scan. In this case, there were also no TSM observed in the subsequent MRIs (S3 Table).

**Regression analysis of factors that influence TSM.** To identify the impact of various, independent effects of factors on TSM, a uni- and multivariate regression analysis was performed (Table 2). While age, gender and weight tended to be associated to TSM in the univariate analysis, age remained as the only factor positively associated with TSM in the multivariate analysis (hazard ratio 9.152, p = 0.049). In order to address the multiplicity of MRIs in 9 patients with TSM, we included this parameter in the regression analysis, but it did not show an association in the univariate analysis (hazard ratio 0.66, p = n.s.).

### Image quality in the dynamic contrast phases

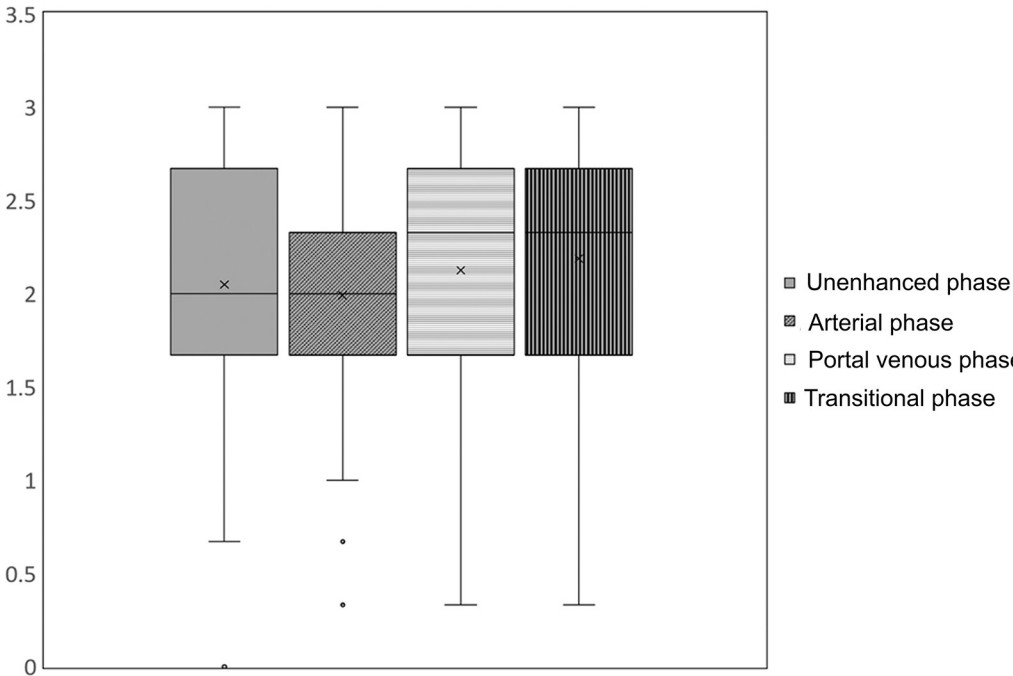

### Image quality in patients with TSM in the dynamic contrast phases

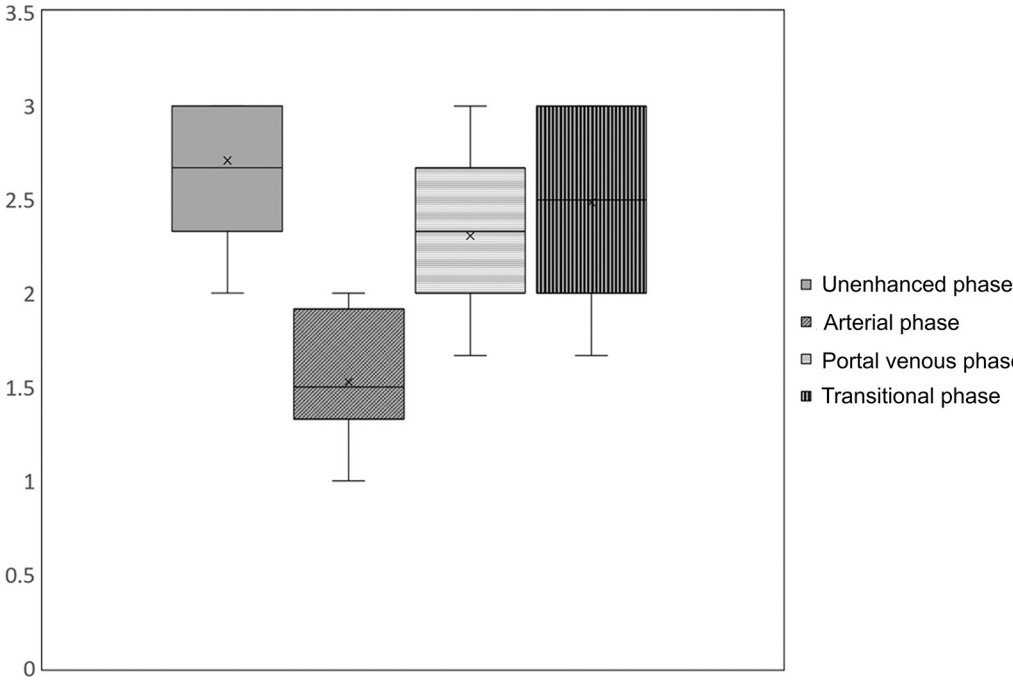

**Fig 2. The image quality in the unenhanced, arterial, portal venous, and transitional phases.** An image quality score of 3 indicated no artifacts, while 0 indicated non-diagnostic images due to artifacts. (2A) The image quality of the study collective and (2B) the image quality of the non-sedated patients with TSM.

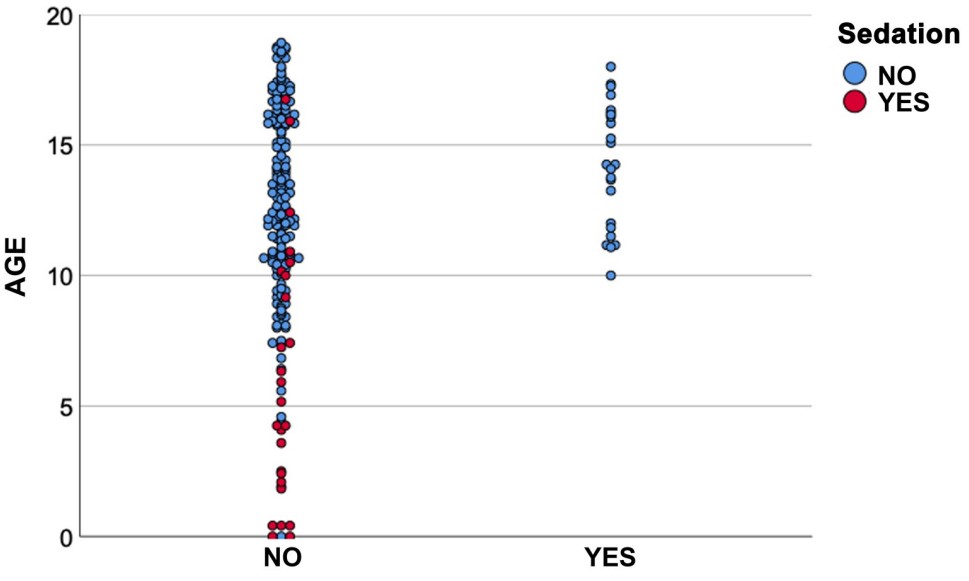

**Fig 3. Relation of age and transient severe motion artifacts in patients with and without sedation.**

## Discussion

In this retrospective single-center study, we observed, for the first time, that transient respiratory motion artifacts in children who underwent Gd-EOB-DTPA-enhanced liver MRI, only occurred after the age of 10 years. After correcting for weight, BMI, sedation, and gender, age remained the only factor associated with TSM. Interestingly, patients under 10 years of age never exhibited signs of TSM, regardless of sedation.

TSM appear as frequently in children as in adults, with a rate of 10.6% in our series, compared to a rate of 10.7–39% in adult patients [3, 18, 19, 21, 29]. TSM can limit the image quality of Gd-EOB-DTPA-enhanced liver MRI in the arterial phase [18–20], which is essential for the differential diagnosis of focal liver lesions [12–14] and for the assessment of the arterial vasculature before and after liver transplantation [15–17]. Due to the increased number of indications for pediatric liver MRI [7, 10, 11, 16], it is crucial to demonstrate that these examinations can be performed with adequate image quality, even in small children.

Two studies have, to date, investigated TSM in pediatric patients. Gilligan et al. reported the experience with Gd-EOB-DTPA-induced TSM in 130 children and compared the presence and severity of TSM in children who were imaged awake versus under general anesthesia [27]. They observed a significantly higher rate of TSM in awake children compared to children under general anesthesia, probably due to a suppression of TSM [27]. In their study cohort, there were significantly heavier (p = 0.0033) and older (p = 0.0066) children in their awake patient group compared to the patients who received anesthesia. Lanier et al. also found TSM in only eight of 102 children (7.84%) without sedation and no TSM in children under sedation, and also discussed the probably protective effect of anesthesia against TSM [28]. These findings were confirmed in our study, with no observed TSM in any of the examinations performed under sedation. Neither of those other studies, however, found any age effect on the occurrence of TSM in non-sedated patients [27, 28], probably because the number of non-sedated patients was too low to draw firm conclusions (63 and 102 patients, respectively). Our study cohort, with 226 cases—199 (88.1%) of them without sedation—is currently the largest

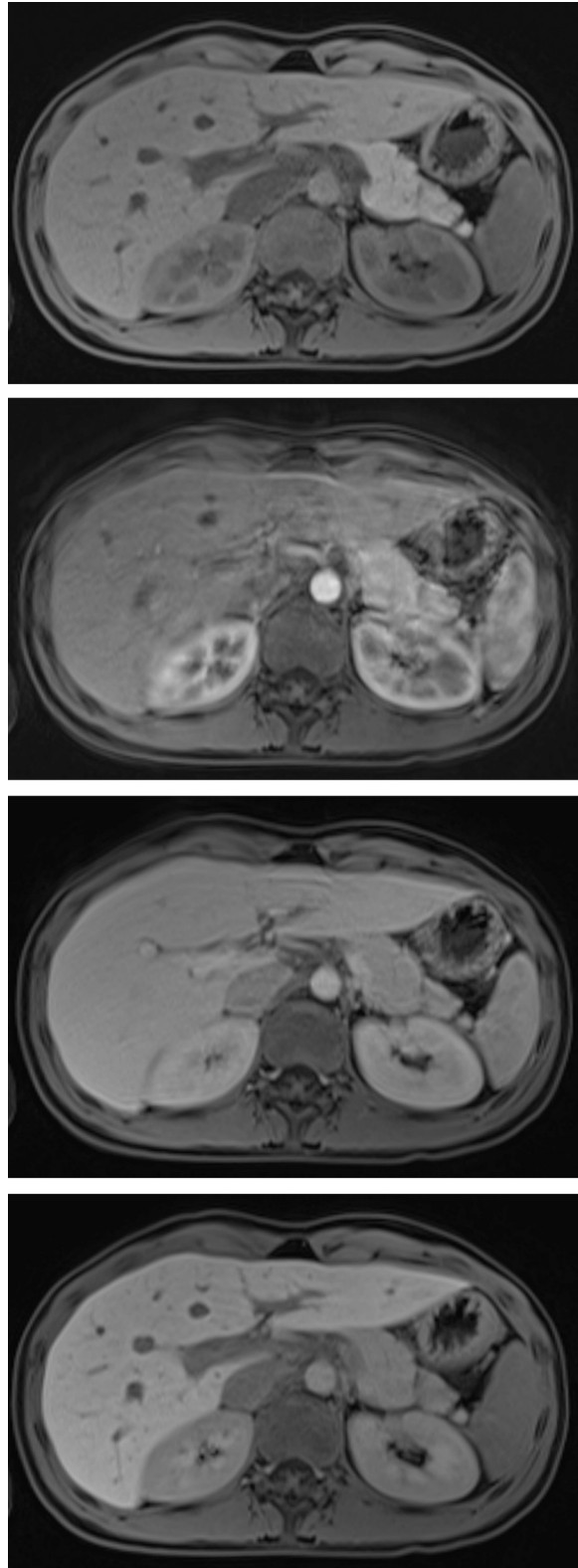

**Fig 4. Example of TSM.** A 16-year-old girl with an alveolar soft-tissue sarcoma of the left thigh who underwent Gd-EOB-DTPA-enhanced liver MRI for a suspected liver metastasis. There were no respiratory artifacts in the unenhanced phase (4A, IQ 3). TSM in the arterial phase (4B, IQ 1). There were still mild motion artifacts in the portal-venous phase (4C, IQ 2), and again, excellent IQ in the transitional phase (4D, IQ 3).

**Table 2. Logistic regression analysis of factors associated with transient severe motion artifacts in the arterial phase.**

| | Univariate | | | | | Multivariate | | | | |
|---|---|---|---|---|---|---|---|---|---|---|
| | B | *p* value | HR | 95% CI | | B | *p* value | HR | 95% CI | |
| Age (≤ 10 years vs. > 10 years) | 2.609 | 0.011 | 13.583 | 1.798 | 102.633 | 2.214 | 0.049 | 9.152 | 1.009 | 82.98 |
| Gender (male vs. female) | 0.912 | 0.041 | 2.490 | 1.040 | 5.960 | | | | | |
| Sedation (non-sedated vs. sedated) | 19.216 | 0.998 | n.a. | n.a. | n.a. | | | | | |
| BMI (kg/m$^2$) | 0.047 | 0.230 | 1.049 | 0.970 | 1.133 | | | | | |
| Weight (kilogram) | 0.017 | 0.062 | 1.017 | 0.999 | 1.035 | | | | | |
| MR repetition (single vs. multiple) | -0.416 | 0.342 | 0.660 | 0.280 | 1.555 | | | | | |

B = unstandardized beta, HR = hazard ratios, CI = confidence interval, n.a., not applicable

cohort to deal with image quality with regard to Gd-EOB-DTPA-induced TSM in pediatric patients, which can only properly be assessed in awake patients. Although there was a clear "protective" trend of sedation towards TSM that was observed in the crude data analysis, there was no association of TSM and sedation using a Fisher´s exact test, p = 0.088, data not shown. Still, we included sedation in the multivariate regression analysis, however it was removed during the first step of the backward elimination process and thus is not independently associated to TSM in our cohort.

The reason for the appearance of TSM only after the age of 10 years is unknown. Due to the off-label use of Gd-EOB-DTPA in pediatric patients, there is little evidence about the nature and occurrence of TSM in this patient population [27, 28]. In adults, pulmonary, cardiac, kidney and liver diseases, gender, body mass index, and history of previous GBCA reactions have been discussed controversially as possible reasons for TSM [20, 21]. A Gd-EOB-DTPA-associated decrease in peripheral capillary oxygen saturation (SpO2) and heart rate have also been discussed [22], as well as breath-hold failure, as the cause of TSM [24].

McQueen et al. described, in 1989, the concept of transient severe hyperventilation upon drug application and concluded that the rapid transient hyperventilation was due to the immediate activation of peripheral chemoreceptors in the carotid body and aortic arch [30]. Polanec et al. speculated that the transiently high concentration of Gd-EOB-DTPA in the initial distribution phase causes a threshold effect that triggers the chemoreceptors centrally, either at the brain stem or carotid artery [3]. If TSM is the result of an activation of chemoreceptors, the reason for the absence of TSM in children younger than 10 years of age could likely be the different regulation or later development of those chemoreceptors in younger children. This was also suggested by Springer et al., who suspected that the maturation of the peripheral chemoreceptors is not complete in childhood, but rather in early adulthood [31].

## Limitations

Our study has several limitations. First, it is a retrospective analysis of a very heterogeneous cohort of pediatric patients who underwent liver MRIs for a variety of reasons. The study cohort included both basically healthy adolescents with, e.g., an incidental benign lesion detected on ultrasound, as well as critically ill children after a liver transplantation, all of which pose different challenges. However, this reflects the clinical reality in a tertiary referral center.

Second, the influence of TSM on the detection of focal liver lesions was not evaluated, and thus, limits the extension of these findings for this part of a liver evaluation. This was due to the fact that many patients did not undergo liver MRI for the assessment of liver lesions, but for hepatobiliary functional or pancreaticobiliary indications, where no focal lesions were present.

Third, the manual injection that was performed in every patient limited the optimal timing of the arterial phase, which would usually not be an issue in adult patients assessed with bolus tracking.

Fourth, the fact that we included several patients, who had repeated MRIs could have led to a clustering effect. There was only one patient, who developed TSM twice within six MRI examinations, and the remaining eight patients had TSM only once, even though up to six examinations were performed. Five of nine patients with repeated MRI, had uneventful subsequent Gd-EOB-DTPA-enhanced MRIs after a TSM event. The rate of TSM in patients with repeated examinations (10 TSM/115 repeated MRIs = 8.7%) was lower than that of the overall cohort (24TSM/226 MRIs = 10.6%); therefore, we think that a clustering bias can be excluded. Furthermore, we included single vs. repeat MR as a covariate in the logistic regression analysis, where it did not prove to be a factor associated to TSM.

## Conclusion

In conclusion, we demonstrated, for the first time, that TSM in the arterial phase of Gd-EOB-DTPA-enhanced liver MRI exclusively appears after the age of 10 years. We speculate that this might be the result of the different respiratory regulation by chemoreceptors between older and younger children. We could also confirm that sedation has a protective effect against the occurrence of TSM, which has been shown previously.

## Supporting information

**S1 Table. Liver MR scan protocol with Gd-EOB-DTPA.**
(DOCX)

**S2 Table. Indications for liver MRI.**
(DOCX)

**S3 Table. Patients with TSM events after i.v. application of Gd-EOB-DTPA-MRI.**
(DOCX)

## Acknowledgments

We would like to thank Thomas Wrba PhD and Susanne Rasoul-Rockenschaub MD, Medical University of Vienna, IT-Systems & Communications, IT4Science, for support in the study data collection.

We would like to gratefully acknowledge Ms. Ines Fötschl for graphical processing of the study images.

We would like to thank Ms. Mary McAllister, M.A., for proofreading the manuscript as a native speaker.

## Author Contributions

**Conceptualization:** Azadeh Hojreh, Dietmar Tamandl.

**Data curation:** Azadeh Hojreh, Christian Lang.

**Formal analysis:** Azadeh Hojreh, Christian Lang, Sarah Poetter-Lang, Dietmar Tamandl.

**Investigation:** Azadeh Hojreh, Christian Lang, Dietmar Tamandl.

**Methodology:** Azadeh Hojreh, Christian Lang, Dietmar Tamandl.

**Project administration:** Azadeh Hojreh, Christian Lang.

**Supervision:** Dietmar Tamandl.

**Validation:** Azadeh Hojreh, Ahmed Ba-Ssalamah, Wolf-Dietrich Huber, Dietmar Tamandl.

**Visualization:** Azadeh Hojreh, Dietmar Tamandl.

**Writing – original draft:** Azadeh Hojreh, Dietmar Tamandl.

**Writing – review & editing:** Azadeh Hojreh, Ahmed Ba-Ssalamah, Christian Lang, Sarah Poetter-Lang, Wolf-Dietrich Huber, Dietmar Tamandl.

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
