## [Decision Letter · Decision Letter 0]

22 Oct 2021

PONE-D-21-16470Gadoxetic acid disodium-induced transient respiratory motion artifacts do not appear in children under the age of 10 yearsPLOS ONE

Dear Dr. Tamandl,

Thank you for submitting your manuscript to PLOS ONE. After careful consideration, we feel that it has merit but does not fully meet PLOS ONE’s publication criteria as it currently stands. Therefore, we invite you to submit a revised version of the manuscript that addresses the points raised during the review process. Please address the comments of Reviewer #1.

We look forward to receiving your revised manuscript.

Kind regards,

Xi Chen

Academic Editor

PLOS ONE

Journal Requirements:

Reviewers' comments:

Reviewer's Responses to Questions

**Comments to the Author**

1. Is the manuscript technically sound, and do the data support the conclusions?

Reviewer #1: Yes

Reviewer #2: No

Reviewer #3: Yes

2. Has the statistical analysis been performed appropriately and rigorously? 

Reviewer #1: N/A

Reviewer #2: No

Reviewer #3: Yes

3. Have the authors made all data underlying the findings in their manuscript fully available?

Reviewer #1: Yes

Reviewer #2: Yes

Reviewer #3: Yes

4. Is the manuscript presented in an intelligible fashion and written in standard English?

Reviewer #1: Yes

Reviewer #2: Yes

Reviewer #3: Yes

5. Review Comments to the Author

Reviewer #1: General comments:

This paper is well written, and the results are clearly presented.

Title:

1. It's better to avoid the title that makes assertions.

Material and methods:

MRI examination protocols

2. Were these MR examinations performed with breath-hold technique? Could young children (e.g. a 0.6-year-old child) hold their breath without sedation?

3. Please show the timing of arterial, portal-venous, and transitional phases.

Statistical analysis

4. How was the normal distribution tested?

5. I think Mann-Whitney U test is commonly used for non-normal distributions. Why the authors used non-parametric t-test?

Reviewer #2: In the manuscript "Gadoxetic acid disodium-induced transient respiratory motion artifacts do not appear in children under the age of 10 years", the authors performed retrospective study on several factors that are potentially associated with transient severe motion. My major concern is the validity of the statistical analysis applied in the study.

As stated in line 199-200: "no TSM were noted in patients who had MRI under sedation", this statement seems to be strongly indicating that the factor of sedation status (non-sedated vs. sedated) may be a very important factor in differentiating the groups with and without TSM. However, such a potentially important factor was discarded in later model selections, and according to the authors' explanation, this is due to the unavailability of hazard ratio in a logistic regression analysis. Such an argument is not quite convincing -- in fact, the absence of TSM events under sedation does not pose any issue to statistical analysis, if more appropriate statistical methods for categorical variables had been employed, such as Fisher's exact test. With such a potentially important factor discarded in the first place, the later analyses become very questionable.

Furthermore, it has come to my attention that the statistical conclusion on the effect of age is only very much marginally significant with p value 0.049, given the multiple other factors and the questionable discarding of the sedation factor, such a weak evidence for two-level factor of age is not likely to yield sufficiently useful and reliable information for future studies.

Reviewer #3: In this manuscript, Azadeh Hojreh et al demonstrated that patients with TSM were significantly older than patients without TSM and TSM were not observed in patients under the age of 10. This result provides us an important speculation that age may have a protective effect against TSM. I think authors well prepared this paper and did steady work. Statistical methods were valid and correctly applied. Interpretation for their results seems appropriate. The references covered the relevant literature.

This manuscript may be accepted for publication.

6. PLOS authors have the option to publish the peer review history of their article (what does this mean?). If published, this will include your full peer review and any attached files.

Reviewer #1: No

Reviewer #2: **Yes: **Daheng He

Reviewer #3: **Yes: **Min Li

---

## [Author Response · Author response to Decision Letter 0]

16 Nov 2021

Reviewer #1: 

General comments:

This paper is well written, and the results are clearly presented.

Query #1. It's better to avoid the title that makes assertions.

Response: 

Thank you for your suggestion. We propose to change the manuscript’s title as follows:

Influence of age on Gadoxetic acid disodium-induced transient respiratory motion artifacts in pediatric liver MRI

Query #2. Were these MR examinations performed with breath-hold technique? Could young children (e.g. a 0.6-year-old child) hold their breath without sedation?

Response: 

In our institution, all non-sedated children do breath-hold-commands exercises with our radiology technologists before they go in the MR-scanner. Therefore, the following of breath-hold-commands depend on the mental development of the child. The sedated children breathe freely during sedation using an oxygen mask, and they are not intubated for MRI, so they generally do not hold their breath. As is already suggested by the nature of this question, many young children are not able to hold breath voluntarily. Thus, we assessed the change in image quality relative to the unenhanced series, which take the individual breathing artifacts into account.

We added an appropriate explanation about our breath-hold-commands during the MR-examination in the section “Patients and clinical data”

Need for sedation was decided by the treating radiologist together with a pediatric anesthesiologist. Children were not intubated, but breathed freely with an oxygen mask. All non-sedated children performed breath-hold-commands exercises with our radiology technologists before they went into the MR-scanner. 

Query #3. Please show the timing of arterial, portal-venous, and transitional phases.

Response: 

The dynamic images were obtained with the same parameters used for the unenhanced sequence with a sequential k-space ordering. The acquisition times were 3 times à 16 seconds, beginning at the time to aortic peak (arterial) and 48 seconds (portal venous), and 300 seconds (equilibrium) (Supporting S1-Table).

We added appropriate details in the Materials and Methods section under “MRI examination protocols”:

The dynamic images were obtained with the same parameters used for the unenhanced sequence with a sequential k-space ordering. The acquisition times were 3 times à 16 seconds, beginning at the time to aortic peak (arterial) and 48 seconds (portal venous), and 300 seconds (equilibrium) (Supporting S1-Table).

Query #4. How was the normal distribution tested?

Response: 

A Levene´s test for similarity of variances was performed and based on the result, either a non-parametric test or a parametric test was used. Levene´s test is an appropriate method to compare distribution between two groups, however, data can still be non-normally distributed despite similarity of variance. Hence, a Mann-Whitney U test was used in case that a normal distribution was not clearly assessable.

Query #5. I think Mann-Whitney U test is commonly used for non-normal distributions. Why the authors used non-parametric t-test?

Response: 

A Mann-Whitney U test was used for non-parametric distributions, e.g. body weight. For normal distributions, a Student´s t-test was used. This was included in the METHODS section:

Variables were compared using a parametric (Student´s t-test) or non-parametric t-test (Mann-Whitney U), or the Chi-square test, where applicable.

Reviewer #2: 

In the manuscript "Gadoxetic acid disodium-induced transient respiratory motion artifacts do not appear in children under the age of 10 years", the authors performed retrospective study on several factors that are potentially associated with transient severe motion. My major concern is the validity of the statistical analysis applied in the study.

Query #1: As stated in line 199-200: "no TSM were noted in patients who had MRI under sedation", this statement seems to be strongly indicating that the factor of sedation status (non-sedated vs. sedated) may be a very important factor in differentiating the groups with and without TSM. However, such a potentially important factor was discarded in later model selections, and according to the authors' explanation, this is due to the unavailability of hazard ratio in a logistic regression analysis. Such an argument is not quite convincing -- in fact, the absence of TSM events under sedation does not pose any issue to statistical analysis, if more appropriate statistical methods for categorical variables had been employed, such as Fisher's exact test. With such a potentially important factor discarded in the first place, the later analyses become very questionable.

Response: 

Thank you for this very important comment. We want to apologize for the unclear formulation, and the misinterpretation this might have caused. Indeed, TSM WAS included in the multivariate analysis. We used a binary logistic regression analysis with backwise elimination using likelihood ratios. In this method, all variables assessed in the univariate analysis are entered into the model and based on the -log2 likelihood ratio when a term is removed, the model continuously removes variables until only variables remain, which exhibit a significant association to the dependent variable. Possible issues with this method would be entering a plethora of variables (leading to an accumulation of random effects, or cross effects) or entering variables with a high degree of collinearity, e.g. weight and BMI. We therefore have included the variables age, gender, sedation, BMI and MR repetition into the model. “Sedation” was removed in the first step of the multivariate backwise elimination model by the software. After running through 3 more steps, “Age” remained as the only covariate associated to TSM. This does not mean that other variables, which are significant in the univariate analysis do not play any role in relation to TSM; it just means, that age is obviously the only variable INDEPENDENTLY associated to TSM, when various other factors are taken into account. 

Regarding the “n.a.” in the univariate analysis of sedation in relation to TSM, the true number that was given by SPSS for the HR was 221550852, which seemed unrealistic to us. We asked our statistician and this could well be due to the fact that zero events were recorded in the sedated subgroup and hence the regression method might be prone to this error (see below). It might also be due to the fact that only about 10% of patients required sedation and that this subgroup was too small to put this observation into the context of the regression analysis.

Therefore, we also performed a crosstable examination, where the Fisher´s exact test yielded a two-sided p of 0.088, which is not shown in the results (please see below, apologies for our german-language version of SPSS). 

Therefore, we feel that after this analysis, the way of interpreting the observation of the multivariate model is appropriate and that TSM and sedation might be related, but not independently. We have added the following paragraph for clarification (methods): 

All variables used in the univariate analysis (except BMI due to the collinearity with weight) were included into a multivariate regression model with stepwise backward elimination.

and (Discussion)

Although there was a clear “protective” trend of sedation towards TSM that was observed in the crude data analysis, there was no association of TSM and sedation using a Fisher´s exact test, p=0.088, data not shown. Still, we included sedation in the multivariate regression analysis, however it was removed during the first step of the backward elimination process and thus is not independently associated to TSM in our cohort.

Query #2: Furthermore, it has come to my attention that the statistical conclusion on the effect of age is only very much marginally significant with p value 0.049, given the multiple other factors and the questionable discarding of the sedation factor, such a weak evidence for two-level factor of age is not likely to yield sufficiently useful and reliable information for future studies.

Response: 

We agree with the reviewer, that a p value of marginally below 0.05 is close to showing a significance. However, this value results after using a multivariate model with heterogeneous input variables in a population with 24 events of the dependent variable. This p-value would be lower, if fewer confounding variables were included in the multivariate analysis, however, we wanted to give a realistic correction for possible confounders. The hazard ratio, however, indicates, that the weighted effect of “age>10 years” on the occurrence on TSM is in fact 13.5 and 9.1 in the uni- and multivariate analysis, respectively. This means, that the older age group has a 9.1-fold increased likelihood of developing TSM, which we do think is clinically relevant.

Reviewer #3: In this manuscript, Azadeh Hojreh et al demonstrated that patients with TSM were significantly older than patients without TSM and TSM were not observed in patients under the age of 10. This result provides us an important speculation that age may have a protective effect against TSM. I think authors well prepared this paper and did steady work. Statistical methods were valid and correctly applied. Interpretation for their results seems appropriate. The references covered the relevant literature.

This manuscript may be accepted for publication.

Response: Thank you very much for your comments.

---

## [Decision Letter · Decision Letter 1]

3 Feb 2022

Influence of age on Gadoxetic acid disodium-induced transient respiratory motion artifacts in pediatric liver MRI

PONE-D-21-16470R1

Dear Dr. Tamandl,

We’re pleased to inform you that your manuscript has been judged scientifically suitable for publication and will be formally accepted for publication once it meets all outstanding technical requirements.

Kind regards,

Xi Chen

Academic Editor

PLOS ONE

Additional Editor Comments (optional):

Reviewers' comments:

Reviewer's Responses to Questions

**Comments to the Author**

1. If the authors have adequately addressed your comments raised in a previous round of review and you feel that this manuscript is now acceptable for publication, you may indicate that here to bypass the “Comments to the Author” section, enter your conflict of interest statement in the “Confidential to Editor” section, and submit your "Accept" recommendation.

Reviewer #1: (No Response)

2. Is the manuscript technically sound, and do the data support the conclusions?

Reviewer #1: Yes

3. Has the statistical analysis been performed appropriately and rigorously? 

Reviewer #1: Yes

4. Have the authors made all data underlying the findings in their manuscript fully available?

Reviewer #1: Yes

5. Is the manuscript presented in an intelligible fashion and written in standard English?

Reviewer #1: Yes

6. Review Comments to the Author

Reviewer #1: The authors have responded to most of the reviewer's concerns and the manuscript has been greatly improved.

> Query #5. I think Mann-Whitney U test is commonly used for non-normal distributions.

> Why the authors used non-parametric t-test?

> Response:

> A Mann-Whitney U test was used for non-parametric distributions, e.g. body weight.

> For normal distributions, a Student’s t-test was used. This was included in the METHODS section:

> Variables were compared using a parametric (Student’s t-test) or non-parametric t-test (Mann-Whitney U), or > the Chi-square test, where applicable.

I've never heard of a “non-parametric t-test”, I think it's a “non-parametric test”.

7. PLOS authors have the option to publish the peer review history of their article (what does this mean?). If published, this will include your full peer review and any attached files.

Reviewer #1: No

---

## [Editor Report · Acceptance letter]

21 Feb 2022

PONE-D-21-16470R1 

Influence of age on Gadoxetic acid disodium-induced transient respiratory motion artifacts in pediatric liver MRI 

Dear Dr. Tamandl:

I'm pleased to inform you that your manuscript has been deemed suitable for publication in PLOS ONE. Congratulations! Your manuscript is now with our production department. 

Kind regards, 

on behalf of

Dr. Xi Chen 

Academic Editor

PLOS ONE